# Challenges and experiences in linking community level reported out-of-pocket health expenditures to health provider recorded health expenditures: Experience from the iHOPE project in Northern Ghana

**Isaiah Awintuen Agorinya**[1,2,3,4]*, **Maxwell Dalaba**[4,5], **Nathan Kumasenu Mensah**[5], **Samuel Tamti Chatio**[6], **Lan My Le**[1,2,7], **Yadeta Dassie Bacha**[8], **Jemima Sumboh**[5], **Gabriela Flores**[9], **Tessa Tan-torres Edejer**[9], **Amanda Ross**[1,3], **Fabrizio Tediosi**[1,3], **James Akazili**[2,5]

1 Swiss Tropical and Public Health Institute, Basel, Switzerland, 2 INDEPTH-Network Secretariat, Accra, Ghana, 3 University of Basel, Basel, Switzerland, 4 University of Health and Allied Sciences, Ho, Ghana, 5 Navrongo Health Research Centre, Navrongo, Ghana, 6 University of Ghana, School of Public Health, Legon, Accra, Ghana, 7 FilaBavi Health and Demographic Surveillance Site, Hanoi, Vietnam, 8 Department of Public Health, College of Health and Medical Sciences, Haramaya University, Harar, Ethiopia, 9 World Health Organization (WHO), Geneva, Switzerland

* iagorinya@gmail.com

**Data Availability Statement:** Data cannot be shared publicly because of the institutions policy

## Abstract

Out of pocket health payment (OOPs) has been identified by the System of Health Accounts (SHA) as the largest source of health care financing in most low and middle-income countries. This means that most low and middle-income countries will rely on user fees and co-payments to generate revenue, rationalize the use of services, contain health systems costs or improve health system efficiency and service quality. However, the accurate measurement of OOPs has been challenged by several limitations which are attributed to both sampling and non-sampling errors when OOPs are estimated from household surveys, the primary source of information in LICs and LMICs. The incorrect measurement of OOP health payments can undermine the credibility of current health spending estimates, an otherwise important indicator for tracking UHC, hence there is the need to address these limitations and improve the measurement of OOPs. In an attempt to improve the measurement of OOPs in surveys, the INDEPTH-Network Household out-of-pocket expenditure project (iHOPE) developed new modules on household health utilization and expenditure by repurposing the existing Ghana Living Standards Survey instrument and validating these new tools with a 'gold standard' (provider data) with the aim of proposing alternative approaches capable of producing reliable data for estimating OOPs in the context of National Health Accounts and for the purpose of monitoring financial protection in health. This paper reports on the challenges and opportunities in using and linking household reported out-of-pocket health expenditures to their corresponding provider records for the purpose of validating household reported out-of-pocket health expenditure in the iHOPE project.

on data sharing. Data are available from the Navrongo Health Research Centre Institutional Data Access /Ethics Committee (contact via https://www.navrongo-hrc.org/content/contact-us) for researchers who meet the criteria for access to confidential data.

**Funding:** This study is part of project that was jointly funded by the INDEPTH-Network in Accra and the Swiss tropical and public health institute of the University Of Basel, Switzerland through a grand from Bill and Melinda gates foundation, grant number OPP1113162. GF, TE, AR were partially supported by WHO.

**Competing interests:** The authors have declared that no competing interests exist.

**Abbreviations:** CHC, Community Health Compounds; CHPS, Community Health and Planning Services; HBS, Household Budget Survey; IDIs, In-depth Interviews; iHOPE, Indepth Household Out-of-pocket Expenditure; LMIC, Low and Middle-income Countries; NHA, National Health Accounts; NHDSS, Navrongo Health and Demographic Surveillance System; NHIS, National Health Insurance Scheme; NHRC, Navrongo Health Research Centre; OOPs, Out-of-Pocket Health Expenditure; OPD, Out-patient Department; SHA, System of Health Accounts; STI, Sexually Transmitted Infection; WHO, World Health Organization.

# Background

Out of pocket health payment has been identified by the System of Health Accounts (SHA) as the largest source of health care financing in low- and middle-income countries [1]. Household out-of-pocket health payments (OOPs) as defined by the system of health accounts 2011 [1] are direct payments for services from household primary income or savings without the involvement of a third-party payer. These payments are usually made by the user at the time of accessing services and includes cost-sharing and informal payments [1]. The over-dependence of health systems on OOPs to finance health care is severe in most low and middle-income countries and implies that health system in these countries will rely on revenue mobilized from households at the time of seeking care. OOPs, can have negative consequences on households' ability to spend on other basic needs in which case they lead to catastrophic health expenditures and, living standards which case lead to impoverishing health expenditures [2,3]. The incidence of catastrophic and impoverishing health expenditures are two strong indicators used to monitor how well a health system is performing in terms of financial protection [4]. These two indicators are solely determined by the extent to which OOPs absorb a household's financial resources [5]. One of the Sustainable Development Goals approved by the United Nations in 2015 (SDG3.8) focuses on health targets including moving towards universal health coverage (UHC). Undoubtedly, OOPs and their negative consequence on households is an important but not exhaustive [4] indicator for tracking progress towards UHC and financial risk protection in low and middle-income countries so an accurate estimate of OOPs in households is critical to the aim of UHC. OOPs incurred by households account for an average of 40% of current health expenditure (CHE) in low-income countries and 30% of current health expenditure in lower-middle-income countries compared to 15–20% in high-income countries [6]. However, the accurate measurement of OOPs has been challenged by several limitations in the sources of data for their estimation [7]. The principal source of these measurement challenges is the tendency for private health care financing to occur without the generation of linked, reliable and comprehensive routine data for national registries, in particular in low and low-middle income countries [8]. In the absence of routine data, these countries rely on national surveys as the main source of data for estimating OOPs [9]. However, these surveys are household-based and have been found to have several limitations due to their design and focus thereby affecting ex-ante post harmonization efforts. Household surveys such as the Living Standard Measurement Surveys (LSMS) have been used extensively in collecting data for estimating current health expenditures and OOPs for most LMIC [9,10]. Some studies [7,11] have attributed the sources of heterogeneity in these surveys to both sampling and non-sampling errors. Unlike sampling errors that are well understood and quantifiable, non-sampling errors result from; survey design, recall period used, the number of questions asked, the choice of the respondent, lack of adequate supervision of primary field staff, tabulations errors among many others [1,7,11] and these errors tend to affect the reliability and comparability of health accounts estimates [10].

To address these limitations and improve the measurement of OOPs in LMIC, there is a need to improve the questionnaires used in these surveys. Establishing a method to generate valid, reliable and comparable information on national and international resource inputs for health is critical for developing policies, managing program implementation and evaluating the efficiency and performance of health systems in developing countries [11]. In the context of improving the measurement of OOPs, the INDEPTH-Network Household out-of-pocket expenditure (iHOPE) project aimed at developing alternative approaches to collect valid and reliable data for the measurement of OOPs in surveys. The main aim of the iHOPE study was to assess the impact of different characteristics such as the specificity, recall period and whether

the survey is door to door or telephone based, on the accuracy of household-reported out-of-pocket payments. Within the framework of the iHOPE project, new modules on household utilization and out-of-pocket health expenditures were developed using the structure of the existing Ghana LSMS (GLSS6) [12]. These new tools were fielded in a cross-sectional study design to collected data on reported health expenditures for cross-validation with provider data ('gold standard') to propose alternative modules which are sensitive to collecting accurate and reliable health expenditure data for estimating OOPs in LMIC. In the context of the iHOPE project, this paper focuses on investigating and documenting the difficulties encountered in the course of implementing the iHOPE study. This particularly identifies the challenges and opportunities in linking household reported health expenditures to their respective provider records for the purpose of validating household reported out-of-pocket health expenditure data in Ghana. This was done using a mixed method approach.

## Methods

### Data source

A total of 2990 households were sampled from the study site using stratified random sampling for the iHOPE Household Budget Survey. Information such as socio-demographic characteristics, general household consumption, and health care utilization & expenditures were collected from all households. Data obtained from the providers include; the name of the patient, the date the health expenditure was incurred, the reason for the visit to the provider and the expenditure incurred by the household for the service/medicine accessed. For the qualitative part of this study, two approaches were employed. For public health facilities where routine health data is collected as part of the administrative process, we reviewed the records and documented the challenges and opportunities in obtaining records of individuals. In private facilities where records are usually not kept, IDIs were conducted in 10 high volume private providers which were identified from the study area for inclusion in the study. They included 3 pharmacy shops and 7 licensed/chemical shops. In-Depth Interviews (IDIs) were conducted with all the sales representatives of these 10 providers. They answered questions on challenges in recording patient data and also provided suggestions to improve the recording process. For the public providers, 8 health centers, 3 clinics and one hospital were included in the study. Each of these types of public/formal providers had a different structure for collecting routine patient information. Patient information such as name, contact address, the reason for visiting the health facility/diagnosis as well as related cost of treatment and/or medicines are routine information kept by these providers for all patients. Data collectors documented all the challenges involved in recording and extracting patient data from these public health care providers for our qualitative analyses. Public providers in this study refer to providers operated by the government of Ghana through the Ghana Health Service and private providers are those operated and managed by private individuals in the community.

**Provider data collections.** Within the Ghana health care system, public health providers who are managed by the government keep patient records as part of routine activities while most private providers either kept minimal transactional records or no records at all. In the first 8 weeks of the project, data collection and retrieval in all private providers was challenged by incomplete and inaccurate patient records since these providers did not routinely collect such data. In the light of these challenges, a brief intervention was carried out to improve data recording in all private providers. A combination of strategies was employed in this brief intervention. The strategies included; deploying a standard template (S1 File) for recording patient data, training private provider owners on how to use it to collect patient information, regular follow-up phone calls to provider owners and regular visits to some providers. The main fields

in the template included; name, address, phone number, referral status, the reason for consultation and cost of treatment/service.

In the case of public providers where patient data collection was routinely recorded, trained field workers were tasked to complete the developed template by reviewing and extracting relevant patient information from the provider record books. The developed template aimed at improving the success of data retrieval from these providers. The criterion for selecting the providers was based on the availability of transactional data (in the case of public providers) or a caretaker who was capable of recording details of the transactions from clients in the case of private providers.

**Matching.** Matched samples in this study refer to households that were linked to their provider records. For any households that reported positive expenditure on any of the health expenditure items, corresponding health provider data was obtained from the provider records using details about the provider obtained from the respondents. The linked household-provider data formed a matched sample used in our analysis. The matching of household and provider data was done at the individual household member level and by spending category. We matched on name, insurance number, date of transaction, household/individual identification number (HDSS ID), name of the facility and the illness/medication the expenditure was incurred.

## Study site

The study was conducted in the Kassena-Nankana East and West Districts of Ghana by the Navrongo Health Research Centre (NHRC). The NHRC operates the Navrongo Health and Demographic Surveillance System (NHDSS) in the two districts. The NHDSS is divided into five (5) zones (Central, North, East, West and South) identified as strata. The estimated population of the districts under continuous demographic surveillance is 152,000. The districts have a rural setting and cover an area of 1,675 square kilometres [13]. The districts have one district referral hospital located in the capital town of the Kassena-Nankana East District (Navrongo) that serves as a referral point for all the health facilities in the districts. The study site has different types of health providers operated and managed by the government and private individuals as seen in Table 1.

## Training of data collectors

University graduates with experience in collecting household survey data and conducting qualitative interviews in demographic surveillance sites were recruited and trained for this study. They were trained on the iHOPE project protocol & survey tools, health provider-patient flow dynamics and how to engage with key persons responsible for running the activities for the providers. A pre-test was conducted at the end of the training session to assess the appropriateness of the data collectors for the work.

## Study design

The study used data from the iHOPE project's Household Budget Survey conducted in Ghana between June 2017 and December 2017. The study was conducted in Navrongo Health and Demographic Surveillance Site (NHDSS) using a cross-sectional design. The iHOPE project aimed at developing alternative tools for estimating OOPs in LMIC. This involved collecting household out-of-pocket health expenditure from sampled households in the study area and cross-validating these expenditures with the corresponding provider. In this paper, we employed a mixed-method approach where both quantitative and qualitative data were used for the analyses. The quantitative part was used to describe partterns and levels of bias between

**Table 1. Characteristics of the study area.**

| | |
|---|---|
| Average km to nearest health facility [1] | 5km |
| Proportion of households with access to cell phones[1] | 72% |
| Number of Health facilities at the HDSS site | 1-Hospital, 1-Health Research Centre, 3-private clinic, 7-health centres, 28-community-based health compounds, 3 Pharmacy shops, 7 high volume chemical shops and Over 50 small chemical/drug sellers, drug peddlers and provision shops |
| Types of Health insurance available at HDSS site | National |
| Health insurance coverage at the HDSS site[1] | 50% |
| Proportion of individuals attending Public health facilities for In-patient cases[1] | 93% |
| Proportion of individuals attending Private health facilities for out-patient cases[1] | 6% |
| Disease classification type in the hospital setting (district hospital) | ICD-10 |
| Recording system in the hospital setting (district hospital) | Paper |
| Recording system In Pharmacy and chemical shops* | Paper |
| In community health centre | Paper |
| In other outpatient care settings | Paper |

source: Computed from unpublished data from the Kassena-Nankana District Health and Management Team (DHMT).

* Data recorded is daily sales.

household OOPs and provider OOPs whilst the qualitative part contexualized the challenges generating the observed biases.

## Data processing and analysis

**Qualitative.**   We conducted qualitative interviews (IDIs) among operators of the private providers to gather data on the challenges and opportunities in collecting the provider data. The IDIs were conducted in English, audio-recorded using digital audio recorders and transcribed verbatim into Microsoft Word. Transcripts were reviewed, and key themes were identified for discussion. A coding list was prepared for data analysis. NVIVO 11 software was used for coding the transcripts and data was analyzed following a deductive content analysis to identify key issues.

Fig 1 shows a flowchart of how data was processed from both public and private health providers for the qualitative analyses.

**Quantitative.**   CSPro 7 was used to capture data for processing and cleaning and then imported into Stata 14 for analysis in the quantitative arm of this study. Descriptive statistics were used to describe socio-demographic characteristics of households, the distribution of types of health providers and the distribution of OOPs spending categories by households. Matching rate was defined in this study as the proportion of individuals from our sampled households whose reported patient details were successfully identified in the records of the corresponding provider where health expenditures were incurred.

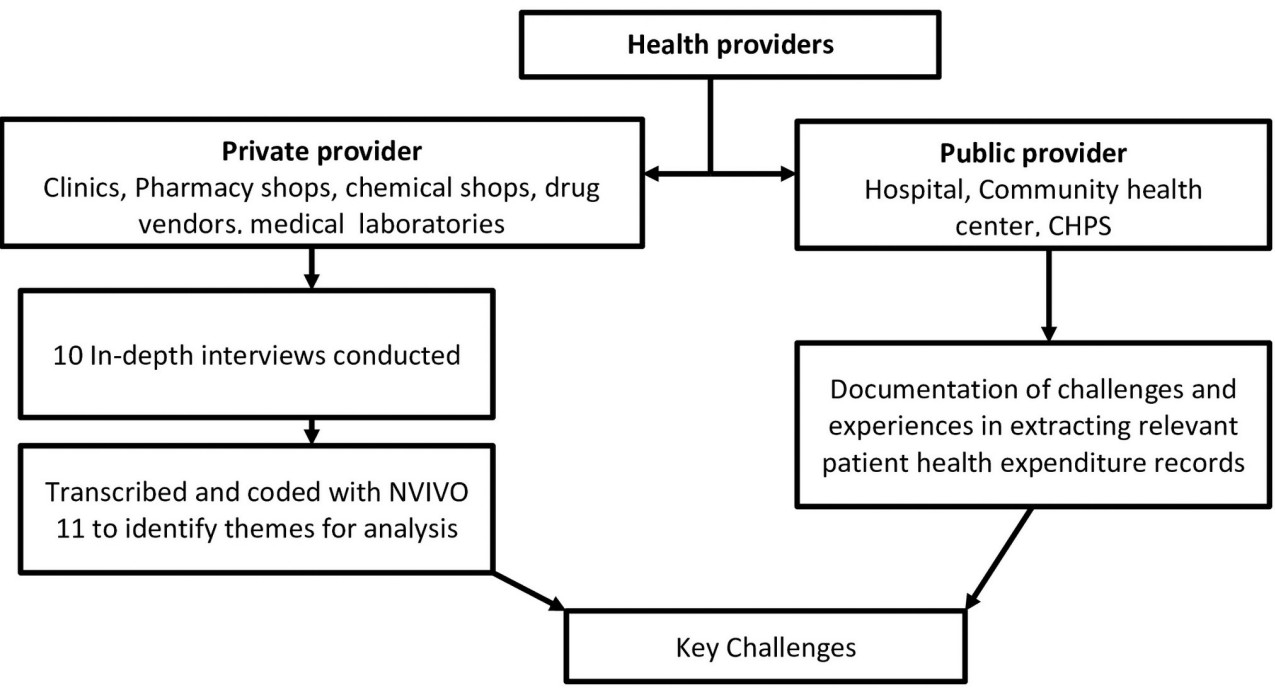

**Fig 1. Structure of data processing for qualitative analysis.**

### Ethics approval and consent to participate

The Ethical Review Board of the Navrongo Health Research Centre, Ghana (NHRCIRB217) approved the conduct of the study. Written informed consent was obtained from adult household heads or an adult person in the household delegated by the household before data collection.

## Results

Generally, our results highlighted very important issues influencing how well community reported data on health expenditure link with corresponding records at health providers in a rural setting. Our results are grouped into 3 parts; 1. Descriptive statistics, 2. Challenges from both patient and providers and 3. proposed solutions to the challenges. First, we present quantitative results of household characteristics, distribution of providers and health care utilization, and their corresponding proportions linked with households. Second, we analyze the challenges from patients and providers influencing the quality of health care utilization and expenditure data. Lastly, we present suggested solutions by the provider owners on how to improve and enhance the recording and quality of health care utilization and expenditure data.

### Socio-demographic characteristics of households

A total of 1402 individuals from 868 households accessed care from different kind of health care providers in this study. Table 2 summarizes the socio-demographic characteristics of the households in our study area. Most households (66.6%) were headed by men, and 66.4% of the household heads were married. The majority of the households (61.3%) were headed by adults between the ages of 35–64 years. More than half of the household heads (53.3%) did not have

**Table 2. Demographic characteristics of household heads.**

| Household characteristics | N | % |
|---|---|---|
| **Sex** | | |
| Male | 578 | 66.6 |
| Female | 290 | 33.4 |
| **Marital Status** | | |
| Not married | 292 | 33.6 |
| Married | 576 | 66.4 |
| **Educational level** | | |
| No education | 463 | 53.3 |
| Primary | 166 | 19.1 |
| Junior High School | 137 | 15.8 |
| Senior High School | 45 | 5.2 |
| Tertiary | 57 | 6.6 |
| **Religion** | | |
| Christians | 452 | 52.1 |
| Islam | 85 | 9.8 |
| Traditional | 281 | 32.4 |
| No religion | 50 | 5.8 |
| **Age group** | | |
| 15–19 | 39 | 4.5 |
| 20–34 | 63 | 7.3 |
| 35–64 | 532 | 61.3 |
| 65 + | 234 | 27.9 |
| **Household size** | | |
| 1 person | 43 | 4.9 |
| 2–5 persons | 397 | 45.7 |
| 6 and above | 428 | 49.3 |

any formal education and about 52% were Christians. The average household size was 6 (3 SD) with about 49.3% of the households having more than six household members.

## Healthcare utilization and matching rates of provider and household information

Table 3 summarizes the distribution of the first visit by household members to a different type of health providers. About 32% of all participants visited the hospital first to seek care, 28% sought first care in community health centers/CHPS, 25% sought first care from licensed Chemical/Pharmacy shops whilst about 8% sought care from unlicensed drug sellers in the community. For each type of provider, the proportion of household that was correctly linked with the provider records varied considerably. For hospital settings, 47% of clients were correctly identified and matched, CHPS recorded 90%, Chemical shops recorded 71%, Diagnostic laboratories recorded 83% and Pharmacy shops 74% (Table 3).

Table 4 shows the distribution of household OOPs to provider OOPs matching rates before intervention and after the intervention. After intervening to improve the recording of OOPs in private health providers and retrieval of records in public providers, we observed an improvement in the matching rates across all spending categories except out-patient services. As shown in Table 4, less than half (47%) of clients visiting a provider for inpatient care could

**Table 3. Distribution of type of providers visited by individuals.**

| Type of provider | Total number of clients attending different provider | the proportion of clients attending different provider | proportion of clients with linked records to household |
|---|---|---|---|
| **Public providers** | | | |
| Hospital | 453 | 32.3 | 46.8 |
| Community Health Centre | 195 | 13.9 | 55.4 |
| CHPS | 196 | 14.0 | 90.3 |
| **Private providers** | | | |
| Chemical Shop | 194 | 13.8 | 71.1 |
| Clinic | 58 | 4.1 | 27.6 |
| Diagnostic laboratory | 29 | 2.1 | 82.8 |
| Hawker/Vendor/Mobile Van | 25 | 1.8 | 0.0 |
| General local shop | 81 | 5.8 | 33.3 |
| Other | 16 | 1.1 | 12.5 |
| Pharmacy | 155 | 11.1 | 73.6 |
| Total | 1402 | | 59 |

be linked to their respective household records before intervention and the proportion increased to 63% after the intervention. Similarly, the matching rate increased from 62% to 83% for medicines, 69% to 73% for preventive care services and 14% to 71% for medical products. Overall matching rate significantly increased from 59% to 77% after the intervention to improve the recording and retrieval of OOPs related patient details at both private and public health providers within the study area.

## Challenges influencing quality and completeness of data recorded by health care providers

The section presents results from interactions with health providers about the factors that potentially influence the quality and completeness of patient data at health care providers in the context of collecting out-of-pocket health expenditure data. The results from the IDIs are structured in three parts. The first part focuses on challenges that relate to patients attending private providers, the second part presents challenges relating to providers in both public and private settings and the third part of this section presents some suggested solutions by health care providers (both private and public) on how to improve patient data quality and completeness.

**Table 4. Proportion of clients at provider correctly linked with household information before and after intervention and by type of service provided.**

| Spending category | Before interventions | | | After interventions | | |
|---|---|---|---|---|---|---|
| | Total number of cases | Number of cases matched | Proportion of cases matched (95% CI) | Total number of cases | Number of cases matched | Proportion of cases matched |
| Inpatient care | 339 | 159 | 47 (41.5–52.4) | 221 | 139 | 63 (56.2–69.3) |
| Out-patient | 551 | 351 | 63 (58.8–67.0) | 53 | 34 | 64 (49.8–76.9) |
| Medicines | 468 | 286 | 62 (57.4–66.4) | 579 | 482 | 83 (79.8–86.0) |
| Preventive care | 32 | 22 | 69 (44.9–83.9) | 60 | 44 | 73 (60.3–83.9) |
| Medical products | 7 | 1 | 14 (0.36–57.9) | 7 | 5 | 71 (29.0–96.3) |
| Total | 1397 | 820 | 59 (56.4–61.) | 921 | 705 | 77 (74.1–79.7) |

**Patient challenges.** *Unwillingness of clients to provide information on "stigmatized" conditions.* It came out strongly across all respondents that some clients were unwilling to provide their details, especially when buying medicines for "stigmatized" or confidential conditions. For example, respondents mentioned that conditions such as diabetes, sexually transmitted diseases (HIV/AIDS, and Gonorrhea), and medicines such as contraceptive pills and aphrodisiacs are stigmatized conditions/medications in the district and for that matter, clients are usually not comfortable to answer questions when they are buying such medicines. They reported that clients usually want to buy the medicine and quickly leave the counter. Respondents mentioned that such clients either refuse to provide the information, provide wrong contact information or lie that they were sent to buy the medicine.

> "*Yeah, there are some types of medicines and cases that people want to keep secret, so when they come here, they don't want to disclose their diseases and they don't want you to even know why they are buying the drug. sometimes they will even tell you they are buying the medicines for someone because they don't want you to know they have such an illness*
>
> (*IDI-In charge, Pharmacy shop*)
>
> . . .for *some clients, their problem is related to the medicines especially the girls, when they come to buy pregnancy test kits and you ask them to write their names, they do not agree for you to record their names because they think that maybe when someone else comes to buy, you will show the name that he/she came to buy pregnancy test kits here*
>
> (*IDI-In charge, Chemical shop*).

It came out from the interviews that sometimes people self-prescribe and consume medicines that may not be needed. In these cases, they are usually uncomfortable when they are being asked questions about what they are using the drugs for. They see the providers to be inquisitive when they are asked about the medicines and contact information. For instance, the youth in the district abuse the use of tramadol tablets (a narcotic pain reliever) with the assertion that tramadol makes them "high" or hyper. Because of this, they are usually not willing to respond appropriately to the providers. For example, a respondent said:

> ". . .for others, *the medicines that they are buying they don't want you to know what they are using it for. The youth these days abuse tramadol a lot and are usually skeptical when you start asking them questions about the medicines. They will simply tell you they are in hurry and leave*
>
> (*IDI-In charge, Chemical shop*)

*Willingness of clients to provide information when buying medicine on credit.* Another issue that came out in the interviews relates to confidentiality when buying medicine on credit. Sometimes, people do not have money when they are sick but would still want to seek treatment. In light of this, some people purchase medicines and payback when they have the money at a later time. Respondents mentioned that some of the clients were not comfortable having their details recorded when they are buying the medicines on credit. These clients perceive this would indicate a lack of trust that they will repay the money owed. On the other hand, some clients perceived that it is only when someone is buying medicine on credit that records are taken. In either case, respondents mentioned that some clients are usually reluctant to give out information.

A respondent for instance said:

"*like those who will come to buy on credit, when they come and you ask them to give you their names they will think that you want to write their names because they are buying on credit and you are taking all this information to be able to trace them when they do not come back to pay. So, they are usually not comfortable giving the information. You need to explain several times. Some will agree but others will still refuse*"

(*IDI-In charge, Chemical shop*)

Similarly, another provider said:

"*Last time one guy came and I asked him to give me his name and he refused. He said that he is not buying the medicine on credit so why do I need his contact information. That I do not have to record his name in my credit record book. I explained that the information was only for record-keeping, but he still said no*"

(*IDI-In charge, Chemical Shop*)

*Limited knowledge about patient details*. It came out of the interviews that sometimes providers cannot obtain the needed information of the patient because some of the buyers do not know the details (Name of patient, home address, phone number) of the patient who requires the medication. Thus, provider operators reported that, sometimes, patients send other persons to buy the medicines on their behalf as such some of these persons are unable to provide full details of the patient to the providers.

"*Sometimes it will just be a small child that will come or just a person sent by a patient. When you ask such as person to provide any information he/she will tell you that they were sent by someone and as such do not know the details.*

"*But if the person has a prescription from the hospital, we are able to record the details from the prescription note. Some will say they were at the hospital premises and they prescribe medicine for someone and the person says he cannot get to the pharmacy because he/she is ill or does not have means of transport that is why I offered to bring the prescription here to assist him/her buy the medicines. So, in these cases, they are unable to provide the needed information about the patient*"

(*IDI-In charge, Pharmacy shop*)

*Clients perceived the process as a waste of time and unnecessary*. In times of ill health, patients are usually in hurry to buy medicines to treat their conditions. It was mentioned across the respondents that some clients complained that they were busy or are not well and for that matter have no time to answer questions. Some clients refused outright while some provided incomplete information and moved away.

"*. . .Others too say it is waste of their time*

(*IDI-In charge, Chemical Shop*)

"*You are wasting his time, for he is in hurry to go and you are asking all sorts of questions . . .*

(*IDI-In charge, Pharmacy shop*)

In addition, given that the patient data recording is not a routine activity and also because patients do not directly benefit from providing their data to providers, they see the process as unnecessary and therefore are not motivated to provide any information to the providers.

*. . . some of the people are difficult, they will ask you whether you are going to give them discount or reimburse them. They say it is unnecessary and a waste of time*

(*IDI-In charge, Chemical Shop*).

*Clients have no trust in the use of their details*. Respondents mentioned that despite the explanations they provide to clients on the reasons for collecting the information, some clients were not still sure what the data was going to be used for and therefore refuse to provide the information.

"*Some understand, but some do not know why you need to know details about them, I need my medicines, am in a hurry to go*

(*IDI-In charge, Pharmacy shop*).

"*for some clients, when you ask them of their Names, they reply by asking, why do you want to know my name, I'm buying medicines from you and you are asking for my name. they don't know why you want to record their names. . .*

(*IDI-In charge, Chemical Shop*).

**Provider challenges.** *Quality of information provided by patients*. In public health providers, every client/patient visiting a health facility is required to provide personal information for recording. Information such as; name, home address, phone numbers, address & contact information and insurance status are obtained from patients. But most of these patients sometimes provide inaccurate information, partly due to memory challenges which makes it a challenge to identify them at a later time. Personal identifying patient records were mostly either incomplete or inaccurate and this posed a huge challenge in retrieving patient records to link with household reported OOPs. The use of a standard template for data retrieval from these providers significantly improved the retrieval of patient records for linkage.

*Providers fear of loss of clients*. Given that providers do not collect patient information such as names and contact address, providers feared that they will lose customers when they continue to ask these questions. They felt collecting the data was more of an intrusion and waste of customer time and are likely to lose these clients if they kept asking for details on their purchases.

"*Sometimes the clients do not see it necessary to provide contact information particularly on sensitive illness or contraceptives. So, they will try to avoid your shop*"

(*IDI-In charge, Pharmacy shop*)

*Provider forget to record data*. Respondents mentioned that they sometimes forget to record patient information in the record books particularly the early days of our study. This is because the recording was not part of the routine activity as well as workload, so they sometimes forget to collect patient information when they come to the shop to buy medicine.

"*sometimes we do forget, the patients will leave before we will realize that we have to take the details of the client*

(*IDI-In charge, Pharmacy shop*)

*Lack of motivation for provider*. Lack of motivation for the salespersons of the chemical shop was cited as a reason for non–recording of patient information. Given that the NHRC has been working with some of these providers and has established that rapport, some respondents were shy to mention that they needed compensation to motivate them to collect patient information. That notwithstanding, few respondents mentioned that motivation to them in the form of money or anything will somehow motivate them to work hard to collect the needed information. Some respondents also explained that sometimes they needed an additional person to help in recording the information and that person needs to be paid for work done.

"*Of course, financial motivation will compel us to try hard to collect the information*"

(*IDI-In charge, Chemical Shop*)

"*It means I have to add more staff and if I am to record this data it means I must pay another person to record the details*"

(*IDI-In charge, Pharmacy shop*)

*Workload to provider*. Most respondents mentioned that they sometimes do not record patient information due to workload, especially busy days such as market days. They stated that sometimes, it is only one person serving at the shop and will not be able to record information of all the customers.

"*Difficulties in recording is because of time. We are few staff here*

(*IDI-In charge, Pharmacy shop*)

"*For my side, non-market days are always better but when it comes to market days where we receive a lot of clients, the clients are always too many that it is difficult to sell and record patient information at the same time. This causes delays in the queue and some clients do get annoyed and go away. . .*

(*IDI-In charge, Chemical Shop*)

Also, some of the workloads relate to double entry. Respondents reported that they had to record in their daily sales book as well as the recording template developed for the iHOPE study. It was easier for providers to record daily sales because only the name of medicine sold and the corresponding amount is recorded.

"*Because I work alone in the shop, I can't record into my daily sales book and on another patient record book at the same time.*

(*IDI-In charge, Chemical Shop)*"

"*it is the pressure. I have to complete my sales books and also your book. It is double work.*

(*IDI-In charge, Chemical Shop*).

**Suggested solutions by providers to mitigate challenges.**  *General education*. Given that recording patient details is not the norm, continues education on the importance of collecting

patient records will improve compliance. They also suggested that the education can be in the form of posters at the chemical shops for clients to read.

"...people should be aware that when they come to the provider, they will be asked questions before medicines are dispensed ...

(*IDI-In charge, Pharmacy shop*)

"...I think it will be better you get a poster and paste it on the wall for those who can read so that when they come, we can show them the poster to read. for those who cannot read we need to continuously educate them verbally...

(*IDI-In charge, Chemical Shop*)

"I think health education should be carried out on air to let people understand that normally when you get into a health facility, and the person is taking your information, you need to have patience and provide the information that is needed. This is going to help the country as a whole since in the future, they can be able to look at the information and tells us the problems or top disease encountered and help us address them"

(*IDI-In charge, Chemical Shop*)

*Client compensation or immediate benefits for clients*. Respondents mentioned that if clients are being compensated for the time, it will motivate them to have time to provide the needed information.

"*If we are compensated, it will certainly improve recording and if the clients know that they will get something, money or any product or package that will help with their health issues, they will be willing to provide information to us*"

(*IDI-In charge, Pharmacy shop*)

*Client follow-up*. Respondents mentioned that if providers make follow up calls or visits to clients to ask about their conditions or the safety of the medicine bought it will go a long way to improve willingness to provide details.

"*some people complained that we collected their details but they did not receive any call from us to find out how they are doing and yet when they visit again we are still asking same questions... I think after taking the contact information, providers should sometimes make some follow up calls to ask about the effect of the medicine on the health of their clients. That will encourage people to provide contact information*"

(*IDI-In charge, Chemical shop*)

*Additional staff at the providers*. For most private health providers, respondents mentioned that given the workload, employing additional staff will ease the workload, workflow and improve recording.

"...to me, when you bring somebody here to sit and collect the data it will help a lot. That person's sole responsibility will be to collect that information and will have the skills to convince people to provide the needed information" (IDI-In charge, Pharmacy shop) "I have been able to record just a few patient details, but the fact is that normally when you are alone in the shop and the clients begin to come, your attention will be on how to serve quickly so that you

*can attend to everyone. In such cases, I think we need to be two in the shop so that one will be writing, and the other will be interviewing and dispensing*"

(*IDI-In charge, Chemical shop*)

*Introduction of computer-based recording system at providers.* Few of the respondents mentioned the introduction of a computer-based recording system as a suggestion to improve patient data capturing. They reported that the computer system will help speed data capturing and avoid repeatedly asking for contact information any time the person comes to the shop to buy medicine. For instance, with preloaded medicines, they do not need to waste time to write the names of the medicines. Also, there will be no need for the provider to ask contact information of a buyer after the first contact information has been captured during the first visit given that it will be stored in the system.

"*I learnt that there are computers that you can enter all the names of the drugs that are here so that when the person comes, you ask the person's name and where the person is from and you just click on the drug. You do not need to waste time to type the drugs*

(*IDI-In charge, Chemical shop*)

"*If our computers are well designed, when we collect the first contact information of a client, we do not need to continue to bore that person about contact information any time he/she comes here to buy drugs again*"

(*IDI-In charge, Pharmacy shop*)

*Monetary motivation to providers.* Very few respondents mentioned that monetary incentive will have motivated them in collecting the additional information since it is additional work for them.

"*Oh, hmm, if we get some money it will motivate us to collect the information. You know collecting that information is not easy. Some people are very difficult, and you need to talk a lot to convince them*"

(*IDI-In charge, Chemical shop*)

## Discussion

There has not been any published study on linking community reported health expenditures directly to corresponding records of providers in any survey. This study presents evidence on the extent to which it is possible to directly link community reported health expenditures to health provider records. Using a mixed-method approach, we present evidence on the degree to which reported health expenditures can be successfully linked with corresponding provider records before and after improving the recording process of patient records in both public and private health providers. The quantitative part of the study focused on describing the distribution of household characteristics and summaries of marching rates across different type of health providers and the accompanying services. Matching rates were obtained for expenditure data in both the private and public providers before and after the intervention to improve provider data recording. The qualitative part of the study was employed to further identify factors that influence the degree of linkage and suggest ways to improve provider data recording in both public and private providers. The discussion is structured along these themes; 1.

Matching rates before the intervention, 2. Matching rates after intervention 3. Challenges in recording and extracting patient information in both public and private providers and 4. Suggested recommendations to improve provider data collection.

Before the intervention to improve data recording in providers, the proportion of household information that accurately matched with provider records was generally low in both public and private providers across diverse health care services. Though proportions that accurately linked were comparable between private providers and public health providers, the proportions were much lower for expenditures incurred in Hospitals and community health Centre (CHC) settings than in Community Health and Family Planning Service Compounds (CHPS) within the public health provider space.

It was also observed that the proportion of individuals that accurately matched with provider records in public providers decreased as the level of care increases (i.e from primary to secondary/tertiary). Our study also revealed that the increasing number of services (consultations, laboratory test, dispensary service, purchase of medical products) provided by public health providers as one move from a lower to a higher level of care played a role in the degree to which household expenditure records are accurately matched to health provider records. This is so because separate unlinked records are kept by each unit within the facility thereby posing a challenge in tracking individual expenditures for aggregation for the same episode of care due to the unavailability of a proper linked recording system. In some cases where there are additional expenditures (e.g purchase of medications) incurred by household members in another provider (second provider) away from the first provider visited for the same episode of care, such expenditures are not accounted for because details (e.g name, expenditures) of the second provider is not kept by the first provider.

The challenges were quite different in private providers. Most private providers tend to operate without generating linked and reliable patient records. To achieve the aim of the iHOPE project, private providers (Pharmacy and chemical shops) who did not routinely keep patient records were engaged to begin collecting routine patient details and expenditures using a standard template developed by the iHOPE project. This engagement was done prior to the commencement of the iHOPE project. Prior to the intervention to further improve the recording of patient details in these private providers, matching rates were marginally higher in private providers than in public providers across all type of care and services provided. This was largely attributed to the use of a standard recording template deployed by the iHOPE project to private providers. The template allowed for the collection of the desired patient information. The matching rates were relatively lower in the public providers because these providers had their own systems for routine recording, these systems are not integrated within the provider and therefore retrieval of patient information becomes a challenge. After the intervention to improve patient data recording in the private providers, matching rates substantially increased.

To understand the factors that drive matching rates in both public and private providers, we engaged with providers. During the engagement, several challenges were identified to influence the completeness and accuracy of patient data. Confidentiality of patient data was a major concern expressed by most clients when providers request details. This was particularly related to stigmatized illnesses or complications from illnesses such as diabetes, STI, and family planning devices such as condoms. Stigmatization has been found [14,15] to influence patients' response to the providers. For instance, Sirey et al. found stigmatization as a major barrier to adherence to the antidepressant drug for treatments of mental illness [15] while McDaniel et al in their study assessing patient willingness to reveal health history information revealed that, a significant number of patients provided inaccurate or incomplete information to questions routinely asked on dental health history form [14]. The findings in our study area

fall in a similar context where stigma and shame especially among STI patients stem from prevalent socio-cultural norms since sex has historically been a stigmatized behaviour and as a consequence of STI. Morris et al, argued that sexual stigma combined with the perpetuated notion of individual responsibility for not adopting certain behaviours has made STIs the symbols of irresponsible behavior [16]. Patients who feared to be stigmatized either refused to provide or provided wrong details to the providers about themselves.

The illegal consumption of prescription drugs came up strongly as a factor that influenced the completeness and accuracy of patient data in most private providers. Self-medication is widespread in the study area and some drugs such as tramadol and other prescriptions medicines which require a prescription from a qualified health professional before they can be dispensed have been found to be the reason for the limited or inaccurate patient data in most private providers. Individuals who consume this class of medicines are mostly unwilling to provide any details for recording. Tramadol, a narcotic-like pain relief drug was reported by providers as one of such drugs being consumed illegally by some individuals in the study area. In the first quarter of 2018, Ghana recorded nationwide high levels of tramadol related crime [17–19]. At the time of writing this paper, the Food and drugs Authority in Ghana had closed down some private chemical drug sellers for dispensing tramadol illegally over the counter to individuals without prescription [19]. As a consequence, most private providers expressed fears of losing their customers who were purchasing these non-prescribed medications. As found in this study, Nga et al, found in Vietnam that, private pharmacy shops owners feared losing customers if they stopped dispensing antibiotics to clients without prescription [20]. It is no secret that when clients feel uncomfortable with one provider they tend to seek care from alternative providers. In the light of this phenomenon, most providers subtly ignore to record details of clients who were receiving care or medicines on non-prescribed medications.

Private providers are generally profit-driven and are the major suppliers of pharmaceutical products in the study area. It is a highly competitive industry that gives individuals easy access to all sorts of services and products with or without prescription from a qualified health professional. Due to this, patients who feel uncomfortable with one provider will choose to see another provider to access the same service or product. Since most of these private providers did not routinely collect patient data, forgetfulness, lack of motivation and workload were issues identified in this study that influenced the completeness of patient data among private providers.

For public providers, challenges were generally about the accuracy of the information given out by the patients for provider records. The non-existence of a proper home address system in the study area (typical of a rural setting) limited efforts in patient identification and matching. Though the study was carried out in a demographic surveillance system (DSS) area under the NHRC where household compounds are uniquely identified for the purpose of population surveillance, most people could not remember the identification numbers when asked by health care providers and for some of those who remembered these IDs either misquoted the whole address or missed out some parts of the ID. The alternate option of using phone numbers provided by the patient at the time of recording was not successful because the cell phone numbers were either out of coverage, incomplete or belong to a distant relative who was most unwilling to provide details about the patient without prior consent.

In the context of estimating OOPs and the production of National Health Accounts, it is important to have comparable health expenditure estimates across different sources of data especially in countries where much private health care financing occurs without the generation of linked, reliable and comprehensive routine data. This study has provided some context to understand the challenges that confront any successful collection of routine administrative and expenditure data in both private and public health providers. This study has also offered

some recommendations that can help reduce the impact of the challenges in collecting such data. Some of the recommendations were implemented by the iHOPE study and yielded improvements in the data recording.

To improve the recording of patient data particularly in the private providers, the iHOPE project implemented some strategies (interventions) within its scope of work to mitigate some of the challenges expressed by the providers. The project; 1. Performed weekly monitoring visits to these providers to remind them to collect patient data, 2. providers compensated monthly as motivation and 3. additional staff were recruited to assist some providers in their shops to reduce the workload. The experience in implementing these interventions yielded positive results as the proportions of individuals whose records were accurately identified increased by about twenty-one per cent. However, not all the strategies implemented by the project are practical feasible in an ideal situation. Regular monitoring of provider data collection activities was the most effective and feasible intervention among the three (3) implemented interventions. In a broader sense, most of the challenges would require a broad and more comprehensive approach to holistically address the issues surrounding provider health records as discussed in this paper.

## Conclusion

Accuracy and completeness of documentation on patient personal details in the context of OOPs were found to be the major challenge in linking individuals to their provider data for the purpose of improving the measurement of Oops. Efforts should not only be focused on improving the survey designs and tools for the accurate measurement of OOPs but also, on the factors that drive the availability, reliability and accuracy of these sources of data. These can be improved by improving data recording systems, setting up follow-up and monitoring systems for recording patient data and routine refresher training to health provider operators. This paper has provided in-depth information about health expenditure data recording which is an essential component in NHA for tracking progress towards universal health coverage and we hope that this paper will provide or add up to the sparse literature available for other future studies.

## Supporting information

**S1 File. Health provider data template.**
(DOCX)

## Acknowledgments

The authors wish to thank all the study participants and health facilities for participating in the iHOPE study. We are very grateful to all the field workers of the iHOPE project who helped in the data collection.

## Declarations

**Consent for publication.** Written informed consent was obtained from all patients for the publication of this manuscript.

## Author Contributions

**Conceptualization:** Isaiah Awintuen Agorinya, Maxwell Dalaba, Nathan Kumasenu Mensah, Lan My Le, Yadeta Dassie Bacha, Gabriela Flores, Tessa Tan-torres Edejer, Amanda Ross, Fabrizio Tediosi, James Akazili.

**Data curation:** Isaiah Awintuen Agorinya, Maxwell Dalaba, Nathan Kumasenu Mensah, Samuel Tamti Chatio, Lan My Le, Jemima Sumboh, Tessa Tan-torres Edejer, Fabrizio Tediosi.

**Formal analysis:** Isaiah Awintuen Agorinya, Maxwell Dalaba, Samuel Tamti Chatio, Amanda Ross, Fabrizio Tediosi, James Akazili.

**Investigation:** Isaiah Awintuen Agorinya, Maxwell Dalaba, James Akazili.

**Methodology:** Isaiah Awintuen Agorinya, Maxwell Dalaba, Lan My Le, Yadeta Dassie Bacha, Gabriela Flores, Tessa Tan-torres Edejer, Fabrizio Tediosi, James Akazili.

**Writing – original draft:** Isaiah Awintuen Agorinya, Maxwell Dalaba, Nathan Kumasenu Mensah, Samuel Tamti Chatio, James Akazili.

**Writing – review & editing:** Isaiah Awintuen Agorinya, Maxwell Dalaba, Nathan Kumasenu Mensah, Samuel Tamti Chatio, Lan My Le, Yadeta Dassie Bacha, Jemima Sumboh, Gabriela Flores, Tessa Tan-torres Edejer, Amanda Ross, Fabrizio Tediosi, James Akazili.

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
