## [Decision Letter · Decision Letter 0]

19 Jan 2021

PONE-D-20-34890

Challenges and experiences in linking community level reported out-of-pocket health expenditures to health provider recorded health expenditures: Experience from the iHOPE project in Northern Ghana

PLOS ONE

Dear Dr. Agorinya,

Thank you for submitting your manuscript to PLOS ONE. After careful consideration, we feel that it has merit but does not fully meet PLOS ONE’s publication criteria as it currently stands. Therefore, we invite you to submit a revised version of the manuscript that addresses the points raised during the review process.

We look forward to receiving your revised manuscript.

Kind regards,

Srinivas Goli, Ph.D.

Academic Editor

PLOS ONE

Additional Editor Comments:

Considering points put forward by reviewer 1, I am going with a decision of Major revision. Reviewer 1 points out several serious issues in the paper. If you can address or reply to reviewer comments, we would like to consider a revised version. However, revision eventually not be accepted. It will be sent out for review once again.

2.) Please include additional information regarding the survey or questionnaire used in the study and ensure that you have provided sufficient details that others could replicate the analyses. For instance, if you developed a questionnaire as part of this study and it is not under a copyright more restrictive than CC-BY, please include a copy, in both the original language and English, as Supporting Information, or include a citation if it has been published previously

3.) We note that you have indicated that data from this study are available upon request. PLOS only allows data to be available upon request if there are legal or ethical restrictions on sharing data publicly. For information on unacceptable data access restrictions, please see http://journals.plos.org/plosone/s/data-availability#loc-unacceptable-data-access-restrictions.

4.) Please amend your list of authors on the manuscript to ensure that each author is linked to an affiliation. Authors’ affiliations should reflect the institution where the work was done (if authors moved subsequently, you can also list the new affiliation stating “current affiliation:….” as necessary).

5.) Your ethics statement should only appear in the Methods section of your manuscript. If your ethics statement is written in any section besides the Methods, please move it to the Methods section and delete it from any other section. Please ensure that your ethics statement is included in your manuscript, as the ethics statement entered into the online submission form will not be published alongside your manuscript.

6. We note you have included two tables which you refer in the text of your manuscript, however both are labelled as Table 1. Please ensure that you label each Table by a separate number in the title and also cite the relevant table number in your text; if accepted, production will need this reference to link the reader to each Table. 

Reviewers' comments:

Reviewer's Responses to Questions

**Comments to the Author**

1. Is the manuscript technically sound, and do the data support the conclusions?

Reviewer #1: No

Reviewer #2: Yes

2. Has the statistical analysis been performed appropriately and rigorously? 

Reviewer #1: No

Reviewer #2: Yes

3. Have the authors made all data underlying the findings in their manuscript fully available?

Reviewer #1: Yes

Reviewer #2: Yes

4. Is the manuscript presented in an intelligible fashion and written in standard English?

Reviewer #1: No

Reviewer #2: Yes

5. Review Comments to the Author

Reviewer #1: The manuscript attempts to provide findings from the INDEPTH-Network Household out-of-pocket expenditure (iHOPE) project in Northern Ghana with specific focus on challenges in linking the data on OOP spending reported by households and service providers. The authors used a mixed methods approach to study the extent of discrepancies in the data obtained from the two sources by means of descriptive analysis and to identify the challenges encountered by service providers in the public as well as private sector in recording relevant data by means of in-depth interviews.

I would like to congratulate the authors on undertaking an important piece of work. There is no doubt that the information generated is of considerable value. The manuscript in its current state however, has significant room for improvement.

The background should clearly spell out the objectives of the larger study, laying more emphasis on those addressed in this paper. The methods and results sections could use a major overhaul. These sections leave out a lot of important information which makes it difficult to judge the technical soundness of the study. Specific comments are provided below for each section of the manuscript. The manuscript needs a thorough proof-read to fix language/grammatical errors as well as inconsistencies in the write-up in the various sections.

BACKGROUND

The objectives of the larger study and those addressed in this paper should be spelt out clearly towards the end of the background section. More emphasis should of course be laid on the specific objectives addressed in this paper.

“To address these limitations and improve the measurement of OOPs in LMIC, there is a need to improve the questionnaires used in these surveys. Establishing a method to generate valid, reliable and comparable information on national and international resource inputs for health is critical for developing policies, managing program implementation and evaluating efficiency and performance of health systems in developing countries.”

Please provide the references for the above statements.

METHODS

The authors should consider re-organising the methods section to begin with the data sources and then following on to the study design and finally data analysis. The text under the section on study design could be limited to the description of the study design without describing the data. Study site, training etc. can all form sub-sections under the data sources section.

The section on data sources needs more clarity. Were private clinics included in the qualitative part of the study? They are mentioned in the flow chart (figure 1) but not specified in the write up.

The flow chart (figure 1) also requires some re-work. The number of interviews conducted in the private sector are mentioned but those in the public sector are not specified in the chart. Transcription as well as documentation of challenges and experiences were carried out for both types of service providers. So the arrows need to connect to both. The chart seems confusing at the moment.

“Matching rate was defined in this study as the proportion of individuals from our sampled households whose reported patient details were successfully identified in the records of the corresponding provider where health expenditures were incurred.”

Authors should specify in detail what they mean by patient details. Which details were sought from both the sources and matched? What technique and criteria were used to qualify records as matched? Did they have to match precisely or some degree of variation was allowed? If variation was allowed, to what extent?

In other words, the authors need to clarify what was matched and how was it matched.

The sections on research design and data analysis also leave out information on the comparison between the pre- and post-intervention period which later appears in the result section. Overall the section needs to be re-organised also to indicate the sequence of data collection as well as analysis. Was the first round of data collected from households and service providers? Was this followed by the collection of qualitative information and the eventual implementation of strategies to address challenges identified? Was this followed by a final round of data collection?

RESULTS

Firstly, the results sections mentions the three broad parts into which the findings have been organised. The sections however, is not organised as such. It would be better to re-organise the results section into three broad sub-sections based on the above description and then further divided into sub-sub-sections within the three main parts.

“Second, we analyse the challenges from patients and providers influencing the quality of health care utilization and expenditure data”

Did the authors mean the challenges described by the patients as well as service providers? Kindly re-phrase for better understanding of the readers.

“Lastly, we present suggested solutions by the provider owners on how to improve and enhance recording and quality of health care utilization and expenditure data”.

Could be re-phrased to say “we present the solutions proposed by the public and private service providers during the course of the interviews….”

Socio-demographic characteristics and matching rates

“Most households (66.6%) were headed by men and were married (66.4%)”

The denominator should be specified for the percentage of married men. Did the authors mean 66.4% of the total households or 66.4% of the (66.6%) households headed by men? From table 1 it appears that 66.4% of the total heads of households surveyed were married (regardless of gender). The write-up requires more clarity.

Further, the first row (after the title row) of table 1 should say something like ‘sex/gender of the household head’. Household size should be pushed to the end of the table since all the other parts refer to the household head.

Although the section heading specifies matching rates, no such rates have been mentioned in the write-up. Matching rates have been provided in a separate section and so this section should be renamed Socio-demographic characteristics.

Healthcare utilization and matching rates of provider and household information

Table 2 as well as its description would benefit from overall matching rates for the private sector as a whole and the public sector. Table 3 has two parts- before and after intervention. The authors should provide some discussion of what they mean by intervention as well as describe the results in both parts of the table. What was the time period before and after the intervention? Was there a transition period (presumably there would be one)? If yes, how did the authors deal with it? All this should be clearly spelt out in the methods which currently do not mention it.

“This proportion was about 69% for clients seeking preventive care and about 38% for medical products.”

According to table 3, the matching rate for medical products was only 14% before intervention and 71% after intervention. Not sure what 38% refers to. Kindly re-check. Was a pre- and post-intervention analysis carried out only in the private sector or in both public and private sector? It seems from the discussion that this was done only in the private sector. Please specify clearly in the methods as well as results.

Patient challenges

All sub-sections under this section rely on information obtained from pharmacists alone. Did the authors not obtain any relevant information from other service providers in the private or public sector? A well-rounded discussion on findings from interviewing all stake-holders including, hospital and clinic staff would help make the section more robust.

Limited information about patient from buyer: this sections should be re-named to better reflect the content

Provider challenges

“Respondents mentioned that they sometimes forget to record patient information in the record books particularly the early days of our study”

It would be interesting to know- what changed during the course of the study? Did the respondents just get used to recording or some external factors contributed to the change?

All sub-sections under this section too rely heavily on the information obtained from pharmacists. What about the other stakeholders interviewed?

Suggested solutions by providers to mitigate challenges

All sub-sections under this section too rely heavily on the information obtained from pharmacists. What about the other stakeholders interviewed?

DISCUSSION

“To improve the recording of patient data at the private providers, the iHOPE project implemented some strategies within its scope of work to mitigate some of the challenges expressed by the providers”.

When were these strategies implemented? What was the basis of choosing specific strategies? Were all the strategies implemented at the same time? Was the study able to capture the impact of these strategies as a whole or individually as well? Such information should be provided in the methods section. The results should also be re-written to emphasis the impact of such strategies.

“These interventions yielded positive results as the proportions of individuals whose records where accurately identified increased by about 21%”

How was this number (21%) arrived at? What was the difference for different types of providers? For what type of providers was the difference most and least pronounced? what could be the reasons behind the difference?

CONCLUSION

The conclusion leaves out the impact of the different strategies employed to address some of the challenges.

Reviewer #2: The study is of critical significance for improving the estimation processes of OOP and its findings can guide similar efforts in other countries. Challenges in estimating OOP, that the study has brought out through the paper, one among the outputs that would come out of the study, could provide guidance for adopting similar enquiries in other settings.

The paper could be accepted with minor revisions and needs one more round of language editing. Some specific comments are provided below:

Table 1: "Proportion of individuals attending Private health facilities for out-patient cases" is 6%. It needs to be verified. 6% seems to be very low for OP care! The survey results also present different public private distribution of care. That would also mean that the sample households may not be true representation of the population.

Line 144: "...households were sampled from the study site using stratified random sampling": What are the strata? need elaboration and presentation of results based on the stratifiers used. while presenting results, only means have been presented. it is important to report CI of results.

Are there no private hospital in the survey districts? The survey does not cover any. Needs justification. Presence of a private for profit tertiary care institution changes health seeking behavior significantly, also provider behavior, both in public and private. need justification of such exclusion

Table 1 appears twice, on page 5 and again at p8.

Table 3 is a crucial part of the results section. It needs to be presented and elaborated more systematically. Also statistical significance of the results need to be presented. It seems some facilities were chosen for intervention. Paper need to mention which facilities were chosen, which were not, on what basis selection was made, etc etc. The results needs to be presented based on the disaggregation, rather than aggregate effects.

The discussion section need to focus on the relevance of the study in estimation of OOP and production of NHA. the discussion covers themes like prescription practice, health seeking behavior and provider behavior which has not been dealt adequately. Discussion section need to elaborate on the implications of mismatch on various estimates produced, which could provide readers greater understanding of the results.

Overall, the study is important for various reasons, but the current paper focuses on a specific aspect. It is important that the paper connects to the larger piece and presents the reults more systematically.

6. PLOS authors have the option to publish the peer review history of their article (what does this mean?). If published, this will include your full peer review and any attached files.

Reviewer #1: **Yes: **Aashna Mehta

Reviewer #2: No

---

## [Author Response · Author response to Decision Letter 0]

23 Jun 2021

Dear Editor,

Many thanks for considering our paper for possible publication in your journal. 

We find your suggestions and the reviewers comments to be very insightful and has contributed to improving the quality of our paper.

Please find below a point by point response to the comments raise and attached is the revise manuscript for your attention and possible consideration for publication

RESPONSE TO REVIEWERS

 REVIEWER 1

Reviewer #1: The manuscript attempts to provide findings from the INDEPTH-Network Household out-of-pocket expenditure (iHOPE) project in Northern Ghana with specific focus on challenges in linking the data on OOP spending reported by households and service providers. The authors used a mixed methods approach to study the extent of discrepancies in the data obtained from the two sources by means of descriptive analysis and to identify the challenges encountered by service providers in the public as well as private sector in recording relevant data by means of in-depth interviews.

I would like to congratulate the authors on undertaking an important piece of work. There is no doubt that the information generated is of considerable value. The manuscript in its current state however, has significant room for improvement.

The background should clearly spell out the objectives of the larger study, laying more emphasis on those addressed in this paper. The methods and results sections could use a major overhaul. These sections leave out a lot of important information which makes it difficult to judge the technical soundness of the study. Specific comments are provided below for each section of the manuscript. The manuscript needs a thorough proof-read to fix language/grammatical errors as well as inconsistencies in the write-up in the various sections.

BACKGROUND

COMMENT: The objectives of the larger study and those addressed in this paper should be spelt out clearly towards the end of the background section. More emphasis should of course be laid on the specific objectives addressed in this paper.

RESPONSE: This has now been addressed in the tail end of the background section

COMMENT: “To address these limitations and improve the measurement of OOPs in LMIC, there is a need to improve the questionnaires used in these surveys. Establishing a method to generate valid, reliable and comparable information on national and international resource inputs for health is critical for developing policies, managing program implementation and evaluating efficiency and performance of health systems in developing countries.”

Please provide the references for the above statements.

RESPONSE: Reference has now been provided (Line 109)

METHODS

COMMENT: The authors should consider re-organizing the methods section to begin with the data sources and then following on to the study design and finally data analysis. The text under the section on study design could be limited to the description of the study design without describing the data. Study site, training etc. can all form sub-sections under the data sources section.

The section on data sources needs more clarity. Were private clinics included in the qualitative part of the study? They are mentioned in the flow chart (figure 1) but not specified in the write up.

The flow chart (figure 1) also requires some re-work. The number of interviews conducted in the private sector are mentioned but those in the public sector are not specified in the chart. Transcription as well as documentation of challenges and experiences were carried out for both types of service providers. So the arrows need to connect to both. The chart seems confusing at the moment.

RESPONSE: Data was collected in different forms for private and public providers. In public providers where routine records are kept, we investigated and documented the experiences in extracting patient data for cross-validation with household reported expenditure related information. However, in private providers where there is mostly no routine records/data collection, we conducted IDIs among operators of these private providers to elicit their views in the challenges involved in collecting such patient/client records. The flow charts has now been modified to show clearly the direction of data collection for the two types of providers. This has now been clarified in the data source section of the manuscript (Line 123)

COMMENT: “Matching rate was defined in this study as the proportion of individuals from our sampled households whose reported patient details were successfully identified in the records of the corresponding provider where health expenditures were incurred.”

Authors should specify in detail what they mean by patient details. Which details were sought from both the sources and matched? What technique and criteria were used to qualify records as matched? Did they have to match precisely or some degree of variation was allowed? If variation was allowed, to what extent?

In other words, the authors need to clarify what was matched and how was it matched.

The sections on research design and data analysis also leave out information on the comparison between the pre- and post-intervention period which later appears in the result section. Overall the section needs to be re-organised also to indicate the sequence of data collection as well as analysis. Was the first round of data collected from households and service providers? Was this followed by the collection of qualitative information and the eventual implementation of strategies to address challenges identified? Was this followed by a final round of data collection?

RESPONSE: Matching was done prospectively at the individual level. That is, anytime during the survey a household member indicated or reported to have incurred OOPs, details of such provider where the OOPs was incurred is obtained from the household member and then the provider traced, the records in such provider reviewed and the associated OOPs extracted the from the provider record books. This enabled us to create a matched paired dataset for the analysis. This has now been addressed in the manuscript in line 163

RESULTS

COMMENT: Firstly, the results sections mentions the three broad parts into which the findings have been organized. The sections however, is not organized as such. It would be better to re-organize the results section into three broad sub-sections based on the above description and then further divided into sub-sub-sections within the three main parts.

“Second, we analyze the challenges from patients and providers influencing the quality of health care utilization and expenditure data”

Did the authors mean the challenges described by the patients as well as service providers? Kindly re-phrase for better understanding of the readers.

“Lastly, we present suggested solutions by the provider owners on how to improve and enhance recording and quality of health care utilization and expenditure data”.

Could be re-phrased to say “we present the solutions proposed by the public and private service providers during the course of the interviews….”

RESPONSE: This has now been addressed in the manuscript

Socio-demographic characteristics and matching rates

COMMENT: “Most households (66.6%) were headed by men and were married (66.4%)”

The denominator should be specified for the percentage of married men. Did the authors mean 66.4% of the total households or 66.4% of the (66.6%) households headed by men? From table 1 it appears that 66.4% of the total heads of households surveyed were married (regardless of gender). The write-up requires more clarity.

Further, the first row (after the title row) of table 1 should say something like ‘sex/gender of the household head’. Household size should be pushed to the end of the table since all the other parts refer to the household head.

RESPONSE: This has now been resolved Line 256

 

COMMENT: Although the section heading specifies matching rates, no such rates have been mentioned in the write-up. Matching rates have been provided in a separate section and so this section should be renamed Socio-demographic characteristics.

RESPONSE: This has now been resolved Line 248

COMMENT: Healthcare utilization and matching rates of provider and household information

Table 2 as well as its description would benefit from overall matching rates for the private sector as a whole and the public sector. Table 3 has two parts- before and after intervention. The authors should provide some discussion of what they mean by intervention as well as describe the results in both parts of the table. What was the time period before and after the intervention? Was there a transition period (presumably there would be one)? If yes, how did the authors deal with it? All this should be clearly spelt out in the methods which currently do not mention it.

“This proportion was about 69% for clients seeking preventive care and about 38% for medical products.”

According to table 3, the matching rate for medical products was only 14% before intervention and 71% after intervention. Not sure what 38% refers to. Kindly re-check. Was a pre- and post-intervention analysis carried out only in the private sector or in both public and private sector? It seems from the discussion that this was done only in the private sector. Please specify clearly in the methods as well as results.

RESPONSE: Authors have addressed these concerns in both the methods and results sections of the manuscript. Authors have clarified what intervention was done and have also re-written the description of table 2 and 3 to reflect the results.

Patient challenges

COMMENT: All sub-sections under this section rely on information obtained from pharmacists alone. Did the authors not obtain any relevant information from other service providers in the private or public sector? A well-rounded discussion on findings from interviewing all stake-holders including, hospital and clinic staff would help make the section more robust.

RESPONSE: per the structure and scope of this study, we interviewed owner/care-takers of private health providers (mostly Pharmacy and chemical sellers). Our decision was based on the fact that, patients/clients are not used to providing their details/information to these private providers as is the case with public providers where such details or information is required before care is given. In this context, we conducted IDIs among private providers (pharmacy and chemical provider owners) to under that challenges they face when they ask a patient/client to provider details about their health conditions, and also to understand the challenges they (providers) face in recording information which they did not previously do. The sub-section on Provider challenges however, includes challenges faced by both public and private providers. Line 281 in the manuscript provides clarity now

COMMENT: Limited information about patient from buyer: this sections should be re-named to better reflect the content

RESPONSE: This has been updated to “Limited knowledge about patient details”. See line 339

Provider challenges

COMMENT: “Respondents mentioned that they sometimes forget to record patient information in the record books particularly the early days of our study”

It would be interesting to know- what changed during the course of the study? Did the respondents just get used to recording or some external factors contributed to the change?

RESPONSE: The improvement was largely due to the Intervention (line XX ) implemented by the iHOPE project to improve recording. This was done through training and regular provider monitoring (phone calls and regular visits). However, we cannot rule out the fact that the providers also got used to the process with time and therefore part of improvement can be attributed to that. We have clarified this in the manuscript by including sentences on the intervention in the methods section (Line 144) 

COMMENT: All sub-sections under this section too rely heavily on the information obtained from pharmacists. What about the other stakeholders interviewed? Suggested solutions by providers to mitigate challenges

All sub-sections under this section too rely heavily on the information obtained from pharmacists. What about the other stakeholders interviewed?

RESPONSE: In the Methods section (Line XX), we clearly indicated that, only Private providers (pharmacy and chemical sellers) were interviewed (IDIs). The rest were a documentation of the providers

DISCUSSION

COMMENT: “To improve the recording of patient data at the private providers, the iHOPE project implemented some strategies within its scope of work to mitigate some of the challenges expressed by the providers”.

When were these strategies implemented? What was the basis of choosing specific strategies? Were all the strategies implemented at the same time? Was the study able to capture the impact of these strategies as a whole or individually as well? Such information should be provided in the methods section. The results should also be re-written to emphasis the impact of such strategies.

“These interventions yielded positive results as the proportions of individuals whose records where accurately identified increased by about 21%”

How was this number (21%) arrived at? What was the difference for different types of providers? For what type of providers was the difference most and least pronounced? what could be the reasons behind the difference?

RESPONSE: The strategies employed included; deployment of a standard template for collecting patient/client details, training of private health provider workers on how to record patient details, regular phone calls and monitoring visits to the providers. All these strategies have not been highlighted in the methods section (Line 144) 

CONCLUSION

COMMENT: The conclusion leaves out the impact of the different strategies employed to address some of the challenges.

RESPONSE: This has not been included in the conclusion section

 

REVIEWER 2

Reviewer #2: The study is of critical significance for improving the estimation processes of OOP and its findings can guide similar efforts in other countries. Challenges in estimating OOP, that the study has brought out through the paper, one among the outputs that would come out of the study, could provide guidance for adopting similar enquiries in other settings.

The paper could be accepted with minor revisions and needs one more round of language editing. Some specific comments are provided below:

COMMENT: Table 1: "Proportion of individuals attending Private health facilities for out-patient cases" is 6%. It needs to be verified. 6% seems to be very low for OP care! The survey results also present different public private distribution of care. That would also mean that the sample households may not be true representation of the population.

RESPONSE: In the context of the Ghana health system, Out-patient care services are not offered in Pharmacy and Chemical shops. However, very few Pharmacy shops and some clinics provider such services in the absence of public providers. The 6% out-patient utilization in private providers is expected (source of data is the district health management team).

COMMENT: Line 144: "...households were sampled from the study site using stratified random sampling": What are the strata? need elaboration and presentation of results based on the stratifiers used. while presenting results, only means have been presented. it is important to report CI of results.

RESPONSE: The strata are only geographic zones demarcated by the Navrongo Health and Demographic Surveillance System for operational purposes. 95% CI have now been added to the estimates in Table 4

COMMENT: Are there no private hospital in the survey districts? The survey does not cover any. Needs justification. Presence of a private for profit tertiary care institution changes health seeking behavior significantly, also provider behavior, both in public and private. need justification of such exclusion.

RESPONSE: There are no private health facilities at the tertiary level in the study area as described in the methods sections (Table 1).

COMMENT: Table 1 appears twice, on page 5 and again at p8.

RESPONSE: This has now been resolved

COMMENT: Table 3 is a crucial part of the results section. It needs to be presented and elaborated more systematically. Also statistical significance of the results need to be presented. It seems some facilities were chosen for intervention. Paper need to mention which facilities were chosen, which were not, on what basis selection was made, etc etc. The results needs to be presented based on the disaggregation, rather than aggregate effects.

RESPONSE: 95% CI have now been included in the Table to allow for statistical significance testing. The narration for the table has also been updated. Section xx3 has also been included to make clear which facilities received the intervention.

The discussion section need to focus on the relevance of the study in estimation of OOP and production of NHA. the discussion covers themes like prescription practice, health seeking behavior and provider behavior which has not been dealt adequately. Discussion section need to elaborate on the implications of mismatch on various estimates produced, which could provide readers greater understanding of the results.

RESPONSE: The discussion section has now been updated to reflect the reviewer’s comment

---

## [Decision Letter · Decision Letter 1]

19 Aug 2021

Challenges and experiences in linking community level reported out-of-pocket health expenditures to health provider recorded health expenditures: Experience from the iHOPE project in Northern Ghana

PONE-D-20-34890R1

Dear Dr. Agorinya,

We’re pleased to inform you that your manuscript has been judged scientifically suitable for publication and will be formally accepted for publication once it meets all outstanding technical requirements.

Kind regards,

Srinivas Goli, Ph.D.

Academic Editor

PLOS ONE

Additional Editor Comments (optional):

Considering my own reading of the paper and reviewers opinion, I am recommending this paper for publication in PLOS One with minor revisions as suggested by Reviewer 1.

Reviewers' comments:

Reviewer's Responses to Questions

**Comments to the Author**

1. If the authors have adequately addressed your comments raised in a previous round of review and you feel that this manuscript is now acceptable for publication, you may indicate that here to bypass the “Comments to the Author” section, enter your conflict of interest statement in the “Confidential to Editor” section, and submit your "Accept" recommendation.

Reviewer #1: (No Response)

Reviewer #2: All comments have been addressed

2. Is the manuscript technically sound, and do the data support the conclusions?

Reviewer #1: Partly

Reviewer #2: Yes

3. Has the statistical analysis been performed appropriately and rigorously? 

Reviewer #1: Yes

Reviewer #2: Yes

4. Have the authors made all data underlying the findings in their manuscript fully available?

Reviewer #1: Yes

Reviewer #2: Yes

5. Is the manuscript presented in an intelligible fashion and written in standard English?

Reviewer #1: Yes

Reviewer #2: Yes

6. Review Comments to the Author

Reviewer #1: The manuscript provides findings from the INDEPTH-Network Household out-of-pocket expenditure (iHOPE) project in Northern Ghana with specific focus on challenges in linking the data on OOP spending reported by households and service providers. The authors used a mixed methods approach to study the extent of discrepancies in the data obtained from the two sources by means of descriptive analysis and to identify the challenges encountered by service providers in the public as well as private sector in recording relevant data by means of in-depth interviews.

I would like to congratulate the authors on undertaking an important piece of work. There is no doubt that the information generated is of considerable value. The authors have also been able to address several of the comments provided to them during the first round of review. The manuscript still has some room for improvement before publication. Specific comments are provided below for different sections of the manuscript. The manuscript needs a final proof-read to fix language/grammatical errors.

METHODS

The section requires some consistency checks. It could also be made more concise by eliminating repetitions.

Provider data collections:

“The public health providers selected include; one hospital, one clinic and seven public health centres. For the private health care providers, ten high volume pharmacy and licensed chemical shops met our selection criteria and were selected”.

This part could be removed as it is a repetition. But please check the number of public health centres. In the previous section eight centres were mentioned instead of seven.

Study design:

The aim of the study may not be repeated in this section.

“We obtained quantitative data from the iHOPE cross-sectional Household Budget Survey (HBS) and Household Health Survey (HHS)”.

The above sentence should have been mentioned in data sources. Also, please elaborate how this data was used. The previous sections only talks about qualitative data.

Data processing and analysis

Most of the section is a repetitive and provides the same information as in the previous section other than the information on recording, transcribing, coding and analysis which appears only under the sub-section for private providers. The section would benefit from greater clarity on the data processing and analysis for public sector as well.

“The private providers included in the study include; Pharmacy shops, chemical shops and medical laboratories.”

Repetition. But medical laboratories have been mentioned for the first time here.

Similarly, please carry out a consistency check for figure 1 specifically for the list of health providers.

The section on matching would better fit this section rather than the section on data sources.

RESULT

“Generally, our results highlighted very important issues influencing how well community reported data on health expenditure link with corresponding records at health providers in a rural setting.”

It may be beneficial to the reader to mention rural setting in the objective and methods as well.

“A total of 1402 individuals from 868 (29%) households accessed care from different kind of health care providers in this study.”

What does 29% refer to? Please clarify in the sentence above.

Does table 4 refer to the matching rates among the private sector alone? It is important to clarify in the text as the intervention was targeted at private providers alone.

“Willingness of clients to provide information on “stigmatized” conditions”

The section heading under ‘patient challenges’ could be rephrased as the unwillingness of clients to… Similar re-phrasing may be carried out for other headings section.

The section on ‘Quality of information provided by patients’ under provider challenges appears to refer to patient challenge rather than a provider challenge.

DISCUSSION

“3. Challenges in recording and extracting patient information in both public and private providers”

The findings section does not present the challenges in the public sector. They seem to appear only in the discussion section. Including a short paragraph in the section on findings could be considered.

“The consumption of non-prescription drugs illegally came up strongly as a factor that influenced the completeness and accuracy of patient data in private provider.”

I think the authors mean the consumption of prescription drugs illegally in the absence of a prescription. Maybe re-phrased for clarity.

“Private providers are generally ….. are the major suppliers of pharmaceutical products in the study area….”

Literature supporting the above statement may please be provided.

A discussion on the suggestions to mitigate challenges may be useful. The findings section presents the suggestions provided by the private sector providers regarding the same. It would be beneficial to the study to include the authors’ perspective on the same. It seems that some of the suggestions like compensating the provider was actually implemented during the study. The discussion could include information on whether this was found to be helpful or the improvement in the matching rate in the post-intervention period was mostly on account of other interventions. Also, a suggesting on compensating the clients may not be practically feasible. A commentary of the suggestions from the perspective of the authors of the study would therefore be useful.

Reviewer #2: (No Response)

7. PLOS authors have the option to publish the peer review history of their article (what does this mean?). If published, this will include your full peer review and any attached files.

Reviewer #1: **Yes: **Aashna Mehta

Reviewer #2: **Yes: **Indranil Mukhopadhyay

---

## [Editor Report · Acceptance letter]

27 Aug 2021

PONE-D-20-34890R1 

Challenges and experiences in linking community level reported out-of-pocket health expenditures to health provider recorded health expenditures: Experience from the iHOPE project in Northern Ghana. 

Dear Dr. Agorinya:

I'm pleased to inform you that your manuscript has been deemed suitable for publication in PLOS ONE. Congratulations! Your manuscript is now with our production department. 

Kind regards, 

on behalf of

Dr. Srinivas Goli 

Academic Editor

PLOS ONE